# Integrating Academic and Community Practices in the Management of Colorectal Cancer: The City of Hope Model

**DOI:** 10.3390/jcm9061687

**Published:** 2020-06-02

**Authors:** Misagh Karimi, Chongkai Wang, Bahareh Bahadini, George Hajjar, Marwan Fakih

**Affiliations:** 1Department of Medical Oncology and Therapeutic Research, City of Hope Comprehensive Cancer Center, Duarte, CA 91010, USA; mkarimi@coh.org (M.K.); chowang@coh.org (C.W.); 2Department of Medical Oncology and Hematology, City of Hope National Medical Center, Mission Hills, CA 91345, USA; bbahadini@coh.org (B.B.); ghajjar@coh.org (G.H.)

**Keywords:** colorectal cancer, precisian medicine, academic and community oncology

## Abstract

Colorectal cancer (CRC) management continues to evolve. In metastatic CRC, several clinical and molecular biomarkers are now recommended to guide treatment decisions. Primary tumor location (right versus left) has been shown to predict benefit from anti-epidermal growth factor receptors (EGFRs) in rat sarcoma viral oncogene homologue (*RAS*) and v-raf murine sarcoma viral oncogene homolog B1 (*BRAF*) wild-type patients. Anti-EGFR therapy has not resulted in any benefit in *RAS*-mutated tumors, irrespective of the primary tumor location. *BRAF-V600E* mutations have been associated with poor prognosis and treatment resistance but may benefit from a combination of anti-EGFR therapy and BRAF inhibitors. Human epidermal growth factor receptor 2 (*HER-2*) amplification was recently shown to predict relative resistance to anti-EGFR therapy but a response to dual HER-2 targeting within the *RAS* wild-type population. Finally, the mismatch repair (MMR)-deficient subgroup benefits significantly from immunotherapeutic strategies. In addition to the increasingly complex biomarker landscape in CRC, metastatic CRC remains one of the few malignancies that benefits from metastasectomies, ablative therapies, and regional hepatic treatments. This treatment complexity requires a multi-disciplinary approach to treatment and close collaborations between various stakeholders in large cancer center networks. Here, we describe the City of Hope experience and strategy to enhance colorectal cancer care across its network.

## 1. Introduction

Colorectal cancer is the second most common cause of cancer-related death in the United States, with an annual incidence of 145,000 cases in 2019 and 51,000 deaths according to the American Cancer Society [1]. The life-time risk of developing colorectal cancer in men and women is 4.6% and 4.2%, respectively, according to 2014–2016 data [2]. The five-year survival rate for metastatic disease and regional disease are 14.2% and 71.3%, respectively [3,4]. Therefore, significant progress is still needed, especially in metastatic colorectal cancer settings.

In this report, we will review the latest approaches in the treatment of metastatic disease. We will also explore the City of Hope approach to delivering optimized care with partnership between academic researchers and community clinicians in cancer care.

## 2. Our Path to Precision Medicine in the Treatment of Metastatic CRC

The management of metastatic colorectal cancer must take into consideration sidedness, as well as molecular biomarkers including *RAS*, *BRAF*, *HER-2*, and MMR status. We have previously extensively reviewed this topic [5,6]. In short, it is now well-established that the benefits of anti-EGFR therapy appear to be limited to *RAS* and *BRAF* wild-type tumors that originate from the left colon [7,8,9]. For this group of patients, the addition of anti-EGFR therapy to combination chemotherapy in the first-line setting is better than bevacizumab according to subgroup analyses from two large randomized trials [10,11]. On the other hand, right-sided tumors as well as *RAS* mutated tumors benefit from the addition of bevacizumab to combination first-line chemotherapy. Second-line treatments are typically shaped by first-line treatment decisions and are addressed in our prior reviews [6].

*BRAF-V600E*-mutated colorectal cancers constitute approximately 8% of metastatic colorectal cancers. These tumors are associated with a poor prognosis and relative chemotherapeutic resistance [12,13,14]. Given their aggressive histology, and based on subgroup analyses from the TRIBE clinical trial, we advocate a combination of 5-FU, irinotecan, and oxaliplatin (FOLFOXIRI) with bevacizumab in the first-line treatment for those deemed to be fit enough to tolerate this regimen [15]. Otherwise, the first-line treatment of *BRAF-V600E*-mutated colorectal cancer is typically managed in a similar fashion as that of *RAS*-mutated metastatic colorectal cancer. A ray of hope has finally emerged in the targeted therapy of these *BRAF*-mutated patients. The BEACON trial has recently demonstrated that in the second-line and third-line settings, a combination of a BRAF inhibitor (encorafenib) and EGFR inhibitor (cetuximab) is better than a combination of chemotherapy (irinotecan with or without 5-FU) and cetuximab [16,17]. Encorafenib plus cetuximab is now to be considered a standard second-line therapy in these patients.

HER-2 amplification occurs in 2% of colorectal cancers and is enriched in left-sided and *RAS* and *BRAF* wild-type tumors [18]. These tumors exhibit a relative resistance to anti-EGFR therapy and respond well to lapatinib and trastuzumab or trastuzumab and pertuzumab, based on the HERACLES and MYPATHWAY trials, respectively [19,20]. This has prompted the National Comprehensive Cancer Network (NCCN) to recommend these treatments in the later lines of treatment for this molecular subgroup. The value of anti-EGFR therapy in these patients is still under investigation. We hope that the ongoing SWOG trial (S1613) will finally shed some light on this issue [21].

Finally, MMR deficiency has emerged as a predictive biomarker of response to PD-1 inhibitors, with or without CTLA-4 inhibitors, in colorectal cancer [22,23]. Nivolumab and pembrolizumab have shown remarkably durable responses in the second- and third-line treatment of these patients [24,25]. The addition of ipilimumab to nivolumab appears to enhance the responses and disease control rates, with promising first-line and beyond outcomes being reported from the CHECKMATE 142 trial [26,27]. Therefore, the NCCN has recommended the integration of these agents (monotherapy or combination) in the second-line (and beyond) treatment of MMR-deficient patients. In addition, the NCCN has recommended the consideration of immunotherapy in MMR-deficient frail metastatic colorectal cancer in first-line settings.

In addition to systemic therapies, one must acknowledge an important role for metastasectomies and ablative therapies in colorectal cancer. These should be important considerations for patients with oligometastatic disease—where surgical intervention can result in a curative outcome and/or improved survival [28,29]. Ablative therapies include microwave or radiofrequency ablation as well as stereotactic body radiation therapy. These are typically used in conjunction with or in lieu of surgery in an individualized fashion. Additional regional therapies in patients with liver-only or liver-predominant metastatic disease include radioembolization and hepatic arterial infusion. The discussion around surgery, ablative therapy, and regional therapy is beyond the scope of this manuscript. However, the above highlights the multi-disciplinary needs in the management of metastatic colorectal cancer.

## 3. Integration of Academic and Community Oncology

Achieving the best outcome in patient care has been the long-standing desire at City of Hope. While academicians design and conduct clinical trials, community physicians provide care to most patients and are critical to clinical trial enrollment and the application of standard of care therapy. The optimal partnership requires significant planning and efforts on both sides. A multimodality approach is becoming exceedingly important in the care of colorectal cancer and should be integrated seamlessly across academic and community practices. Here, we describe our efforts to enhance partnerships between our City of Hope Cancer Center and our associated Community Practice Satellites.

### 3.1. Integrating Community Practices in Tumor Board Discussions

Increasingly, the management of colorectal cancer requires the involvement of a multidisciplinary team including medical oncologists, radiation oncologists, pathologists, gastroenterologists, cancer geneticists, colorectal cancer surgeons, thoracic surgeons, and surgical oncologists. At City of Hope, we have conducted Gastrointestinal Oncology Tumor Boards on a twice-weekly basis (Mondays and Thursdays) with representative members from each of the disciplines above. **Tumor boards are disease specific and are run on a weekly basis, with email invitations generated to all interested community physicians. In general, community practices participate when they have an interesting case that requires input in a multidisciplinary setting.**

During these meetings, complex colorectal cases are discussed to determine the best treatment options. The recommendations made span from chemotherapy/immunotherapy/targeted-therapy refinement to decisions regarding metastasectomy, adjuvant therapy, radiation therapy, stereotactic body radiation therapy (SBRT), radioembolization, and hepatic artery infusion therapy. Community practice physicians present their cases remotely to these conferences, providing them with the opportunity to benefit from a multidisciplinary review of their cases and the receipt of a multispecialty input regarding a comprehensive approach towards colorectal cancer. **Such participation has altered treatment management in select cases (such as recommendations regarding adjuvant or neoadjuvant therapy or complex surgery), in allowing to link them to certain cases with appropriate clinical trials in our Duarte campus.**

### 3.2. Integrating Clinical Trials in Community Practices

Clinical trial access has become increasingly important for our colorectal cancer patients. However, the proximity of patients to a main center that provides these treatments has been and remains a main problem that has hindered patient enrollment. City of Hope is supporting a strong initiative to activate clinical trials in our various community centers as part of our mission to enhance research and improve patient access to novel therapeutics. We have partnered with our Community Practices to activate studies of interest to the community, with a strong focus on Phase II and III studies, early-phase investigator-initiated studies, and cooperative group trials. **The end result has been an increase in the accrual rate in community practices, where 70–100 patients have been enrolled in therapeutic clinical trials on a yearly basis.** This has particularly applied to colorectal cancer, where we have activated therapeutic trials that span first-line, second-line, and third-line studies (examples are listed in Table 1).

Potential studies are vetted by community physicians for the feasibility of the associated research procedures and the availability and potential interest of an eligible colorectal cancer population (Figure 1). Only once a study is identified to be fit for a specific community practice is it endorsed for activation in that site. To exemplify, CanStem303C (NCT02753127) was activated through our enterprise to address the value of the cancer stem cell inhibitor BBI-608 in the second-line treatment of metastatic CRC. Out of 25 patients enrolled in this study across four research sites, 13 were enrolled in our main Cancer Center campus, while 12 patients were enrolled through three additional community centers.

## 4. Referring Routine and Complex Cases

The management of colorectal cancer, whether metastatic or localized, is well-characterized and can be performed without significant barriers in community practices. The availability of a large network of providers across a large geographical area allows for easy access to the medical provider, less commuting time, and improved patient satisfaction. Most of the cases seen on our campus can therefore be managed in a more convenient location, which suits patients traveling a long distance from our Cancer Center. These options are discussed with our patients seen on our main Duarte campus, with an appropriate referral made to a more convenient City of Hope Community Practice for continuity of care (Figure 2).

On another note, certain colorectal cancer patients that are treated in our community practices may benefit from complex specialized services that are only feasible in our main campus. These patients are referred for treatment and continuity of care from our satellite practices to our Duarte campus. For example, the use of hepatic arterial infusion pumps for regional chemotherapy in an adjuvant setting after hepatic metastasis resection requires special surgical expertise as well as a specialized supporting team (interventional radiology, nurses trained for pump access and troubleshooting, and oncologists experienced in regional hepatic chemotherapy). We have taken the conscious decision to centralize the management of these patients at our main campus.

## 5. Standardization of Treatment Pathways in Colorectal Cancer

The creation and standardization of treatment pathways are key to the administration of quality care across our network. City of Hope is a National Network Cancer Center Network (NCCN) member. We contribute to various committees in the NCCN and support its published guidelines [30]. However, the guidelines are broad and are not easy to navigate across our sprawling network of community practices. Several years ago, we joined the Via Oncology network (currently ClinicalPath), which provides an easy-to-navigate clinical pathway for medical oncologists across all disease sites. These guidelines strive to reduce variability, focus on cost-effectiveness, and seek patient-friendly (less toxic and easier to administer) regimens. Since these pathways are meant for academic and community practices alike, we have included representative members from our Cancer Center and Community Practices on the Gastrointestinal Cancer Via Oncology Committee. The committee meets on a quarterly basis and discusses recently published or presented clinical data that can impact the recommended Via Oncology treatment guidelines. Every cancer patient treated in our institute is navigated through these pathways electronically, and the data are reviewed to assess treatment adherence and guideline compliance.

## 6. Educational Efforts

Educational programs (certified medical education) across a variety of cancers are hosted on a weekly basis on our Duarte Campus (Cancer Center). All faculty members across our satellite offices have the capability of attending in person or remotely. In addition, our medical oncology department hosts bi-annual symposia for medical oncology. During these meetings, each disease site, including colorectal cancer, is co-hosted by a Cancer Center Academician and a Community Practice physician. Community practice physicians leading such efforts have an existing interest and expertise in the assigned respective area and lead the discussions on the latest standards of care. Such programs increase the interaction between our research faculty and clinical faculty and enhance learning collaborations across our network.

## 7. Conclusions

The close interplay and collaboration between our Cancer Center and our Community Practices is essential to optimize clinical care for colorectal cancers across our community. These collaborations include educational activities, the standardization of treatment pathways, clinical trials, and the cross-referral of patients to address patient convenience and treatment complexity. The constant cross-talk between our Academic and Clinical Faculty across the network insures that the best standards in colorectal cancer are applied across our capture area.

## Figures and Tables

**Figure 1 jcm-09-01687-f001:**
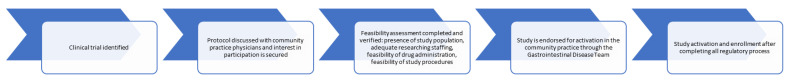
Process for clinical trial activation in a City of Hope Community Practice.

**Figure 2 jcm-09-01687-f002:**
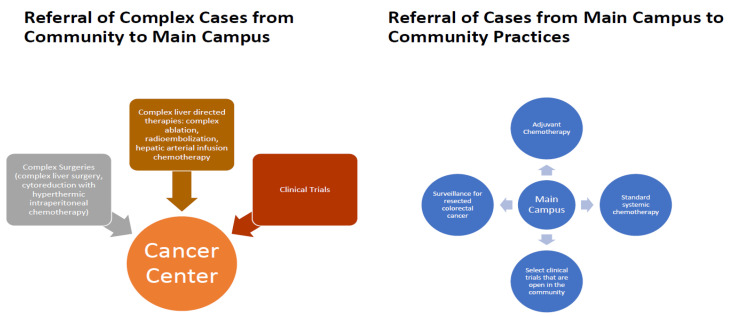
Cross-referral patterns between the City of Hope main cancer center and community practices.

**Table 1 jcm-09-01687-t001:** Selection of colorectal cancer trials activated in City of Hope Community Practices.

NCT Number	Line of Treatment	Title
NCT04094688	First Line	Vitamin D3 With Chemotherapy and Bevacizumab in Treating Patients With Advanced or Metastatic Colorectal Cancer (SOLARIS)
NCT02753127	Second Line	A Study of Napabucasin (BBI-608) in Combination With FOLFIRI in Adult Patients With Previously Treated Metastatic Colorectal Cancer (CanStem303C)
NCT03317119	Third Line	Trametinib and Trifluridine and Tipiracil Hydrochloride in Treating Patients With Colon or Rectal Cancer That is Advanced, Metastatic, or Cannot Be Removed by Surgery

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
