# Peer review of "Integrating Academic and Community Practices in the Management of Colorectal Cancer: The City of Hope Model"

_jcm, 2020, doi:10.3390/jcm9061687_

Round 1

Reviewer 1 Report

General Commentary:

The authors begin by discussing a broad review of updated knowledge of current landmark trials in colorectal cancer treatment.  The authors then go on to detail City of Hope's efforts for incorporating protocols in for improving access to patients in their provider network.  This includes an integrative approach which centers on a multidisciplinary collaboration.  Although these efforts are certainly altruistic, this is a common approach to most tertiary hospitals and Veterans Affairs networks with multidisciplinary cancer centers.  Because of this, the novelty of City of Hope's approach is questionable.  Unfortunately, the authors provide no data on outcomes supporting City of Hope's efforts except for one sentence referring to how many patients are enrolled in clinical trials.  Without this, this manuscript serves more as a brochure for City of Hope than a scientific contribution.  

Additional Comments:

  1. First sentence of the Introduction: "Colorectal cancer is the second most common cancer in the United States with an annual incidence of
    26 145,000 cases in 2019 and 51000 deaths according to Seer data"

Comment: Stating the "second most common cancer" is misleading in that it does not specify which metric.  The authors need to specify if this is referring to incidence, prevalence, etc. as well as which dataset this refers to.  Simply referring to annual incidence data in the second half of the sentence is not the only metric commonality.  Would rephrase this sentence to clarify.  To my knowledge, the American Cancer Society and National Cancer Institute refer to colorectal cancer as the second most common cancer in men/women combined in regard to mortality.  Additionally the NCI SEER registries have different versions (9,13, 18, 21, etc.).  This should be clarified and "Seer" needs to be in all CAPS as it refers to an acronym.  Skin cancer needs to be a caveat added as well such as "x most common cancer after skin."

2. Reference to Colorectal Cancer (CRC)

Comment: CRC is frequently not used throughout the manuscript and thus goes against using the abbreviation.

3. Section 2

Comments: This section spends a significant amount of text detailing landmark trials and instead could be focused on detailing patient outcomes. 

Author Response

Dear Reviewer,

Thank you for your insightful comments. Please see below our response.

General Commentary:

The authors begin by discussing a broad review of updated knowledge of current landmark trials in colorectal cancer treatment.  The authors then go on to detail City of Hope's efforts for incorporating protocols in for improving access to patients in their provider network.  This includes an integrative approach which centers on a multidisciplinary collaboration.  Although these efforts are certainly altruistic, this is a common approach to most tertiary hospitals and Veterans Affairs networks with multidisciplinary cancer centers.  Because of this, the novelty of City of Hope's approach is questionable.  Unfortunately, the authors provide no data on outcomes supporting City of Hope's efforts except for one sentence referring to how many patients are enrolled in clinical trials.  Without this, this manuscript serves more as a brochure for City of Hope than a scientific contribution.  

Additional Comments:

  1. First sentence of the Introduction: "Colorectal cancer is the second most common cancer in the United States with an annual incidence of
    26 145,000 cases in 2019 and 51000 deaths according to Seer data"

Comment: Stating the "second most common cancer" is misleading in that it does not specify which metric.  The authors need to specify if this is referring to incidence, prevalence, etc. as well as which dataset this refers to.  Simply referring to annual incidence data in the second half of the sentence is not the only metric commonality.  Would rephrase this sentence to clarify.  To my knowledge, the American Cancer Society and National Cancer Institute refer to colorectal cancer as the second most common cancer in men/women combined in regard to mortality.  Additionally the NCI SEER registries have different versions (9,13, 18, 21, etc.).  This should be clarified and "Seer" needs to be in all CAPS as it refers to an acronym.  Skin cancer needs to be a caveat added as well such as "x most common cancer after skin."

Response: Revised as suggested. This sentence was rephrased to “Colorectal cancer is the second most common cause of cancer related death  in the United States with an annual incidence of 145,000 cases in 2019 and 51000 deaths according to the American Cancer Society”. Please refer to line 25-26.

We did not claim that our approach is novel. However, we do represent a well-delineated approach between our comprehensive cancer center and community practices. We believe that our approach and experience can be helpful for institutes contemplating an expansion of their clinical activities to the community.

  1. Reference to Colorectal Cancer (CRC)

Comment: CRC is frequently not used throughout the manuscript and thus goes against using the abbreviation.

Response: CRC was used in the tittle and abstract section for word limits. In the main manuscript section, we use colorectal cancer instead of CRC. In the main manuscript section, we have replaced CRC with colorectal cancer

  1. Section 2

Comments: This section spends a significant amount of text detailing landmark trials and instead could be focused on detailing patient outcomes. 

Response: The goal of this manuscript was to detail the interaction between a CCC and its community practices with a focus on colorectal cancer. The focus is not patient outcomes, which would vary dependent on studies and patient population.

Reviewer 2 Report

In their manuscript entitled "Integrating Academic and Community Practices in the Management of Colorectal Cancer: The City of Hope Model", Karimi and colleagues proposed a review of colorectal cancer (CRC) treatment and management and further described how City of Hope network is dealing with these patients.

The article is well written and clear.

As a translational scientist, I would be interested in understanding how patients are screened for the referenced biomarkers. How are the results centralized and communicated to practionners.

Otherwise as minor comments:

Figures' font is too small and need to be increased.

NCCN is not defined at first occurence.

Author Response

Dear Reviewer,

Thank you for your insightful comments. Please see below our response.

Comments and Suggestions for Authors

In their manuscript entitled "Integrating Academic and Community Practices in the Management of Colorectal Cancer: The City of Hope Model", Karimi and colleagues proposed a review of colorectal cancer (CRC) treatment and management and further described how City of Hope network is dealing with these patients.

The article is well written and clear.

As a translational scientist, I would be interested in understanding how patients are screened for the referenced biomarkers. How are the results centralized and communicated to practitioners.

Response:  This is a complex topic that can be the subject of its own article- part of the reasons it was not expanded upon. Ideally, we would love to have a centralized system for molecular testing and a molecular data base. Our institute is currently working on such a model through our precision medicine. We are planning to roll out 8000 tumor NGS assays fully funded by our precision medicine so that every patient has their tumor profiled. The results are minable for alterations associated with active clinical trials in the community and in our clinical practice. This initiative will be initiated in 2020. However, for the time being, we have been dependent on a non-centralized assessment of NGS results that are done through different laboratories and with different panels. The current model requires heavy involvement of clinical research coordinators who mine the data from potentially eligible patients to assess if they match any of our biomarker driven studies.

Otherwise as minor comments:

Figures' font is too small and need to be increased.

Response: Figures’ font has been increased and bolded.

NCCN is not defined at first occurrence.

Response: Added National Comprehensive Cancer Network on line 58-59

Reviewer 3 Report

CRC is a complex disease with various clinical and molecular biomarkers that need to be considered during clinical decision making. Metastatic disease is far more complex with severely diminished outcomes. Therefore, there is a great need to not only implement new treatment options but improve how patient care is managed and clinical decisions are made. The authors advocated for integrating satellite community medical practices and large cancer centers and discuss the current networks in place at City of Hope to optimize this.  

Overall this paper is well written and addresses an important problem facing CRC patient care. City of Hope has implemented complex partnerships with community clinicians and incorporated them into multidisciplinary tumor board discussions and clinical trial recruitment occurring at both the main institute and satellite offices. A few notes to help improve the flow of the paper and impact for the readers.

  1. Layout and headings can be better organized to help the flow for the reader and highlight important aspects. Specifically sections 3.1-3.3

  1. There are several redundancies in the paper, while sometimes these are useful to underline important aspects, a few can be eliminated.

  1. Papers pertaining to sidedness and molecular markers have also been reviewed by others, a few of these should be included.

  1. The main point of this paper is to share the success of the City of Hope model, which is done in general terms, however, specifics are often left out. For other institutions to truly learn from the City of Hope model more specifics are needed. A few examples to include along with others if possible:
    1. Section 3.2 How are community doctors made aware of tumor board opportunities? How many satellite offices participate? Has this inclusion improved patient outcomes and to what degree?
    2. Section 3.3 How has City of Hope “ partnered with our Community Practices to activate studies 102 of interest to the community, with a strong focus on Phase II and III studies, early-phase investigator-103 initiated studies and cooperative group trials.”? How has this affected recruitment?

Figure 2 the left figure seems to say the patients joining all clinical trials are referred to the main cancer center but the text states trials are being implemented at community centers as well as the main campus.

Section 5. This is a critical point in this paper. However no specifics in how these treatment pathways are used for metastatic CRC. Line 140 needs a reference to where someone can locate the published NCCN guidelines.

Conclusion: It is a little thin. Discussions on how the model can be improved and how it will adapt to changes in the future are needed. Additionally, a proposed quantitative metric to assess the success of the model needs to be discussed, even if briefly.

Author Response

Dear Reviewer,

Thank you for your insightful comments. Please see below our response.

Comments and Suggestions for Authors

CRC is a complex disease with various clinical and molecular biomarkers that need to be considered during clinical decision making. Metastatic disease is far more complex with severely diminished outcomes. Therefore, there is a great need to not only implement new treatment options but improve how patient care is managed and clinical decisions are made. The authors advocated for integrating satellite community medical practices and large cancer centers and discuss the current networks in place at City of Hope to optimize this.  

Overall this paper is well written and addresses an important problem facing CRC patient care. City of Hope has implemented complex partnerships with community clinicians and incorporated them into multidisciplinary tumor board discussions and clinical trial recruitment occurring at both the main institute and satellite offices. A few notes to help improve the flow of the paper and impact for the readers.

  1. Layout and headings can be better organized to help the flow for the reader and highlight important aspects. Specifically sections 3.1-3.3

Response: Reorganized. Please refer to line 80, 88, and 101

  1. There are several redundancies in the paper, while sometimes these are useful to underline important aspects, a few can be eliminated.

Response: We are happy to revise or visit the sentences in question if pointed out.

Papers pertaining to sidedness and molecular markers have also been reviewed by others, a few of these should be included.

Response: we have previously extensively reviewed the sidedness as well as molecular biomarkers in the management of colorectal cancer. Here is the reference:

Fakih, M.G. Metastatic Colorectal Cancer: Current State and Future Directions. Journal of Clinical Oncology 2015, 33, 1809-1824, doi:10.1200/jco.2014.59.7633.

Sandhu, J.; Lavingia, V.; Fakih, M. Systemic treatment for metastatic colorectal cancer in the era of precision medicine. Journal of Surgical Oncology 2019, 119, 564-582, doi:10.1002/jso.25421.

In addition, we had a summary about sidedness and molecular biomarkers on line 35-43.

  1. The main point of this paper is to share the success of the City of Hope model, which is done in general terms, however, specifics are often left out. For other institutions to truly learn from the City of Hope model more specifics are needed. A few examples to include along with others if possible:
    1. Section 3.2 How are community doctors made aware of tumor board opportunities? How many satellite offices participate? Has this inclusion improved patient outcomes and to what degree?Tumor boards are disease specific and are run on a weekly basis with email invitations generated to all interested community physicians. In general, community practices participate when they have an interesting case that requires input in a multidisciplinary setting. Such participations have altered treatment management in select cases (such as recommendations regarding adjuvant or neoadjuvant therapy or complex surgery) as allowed to link to certain cases with appropriate clinical trials in our Duarte campus. Please refer to line 93-96; 103-106
    2.  
    3. Response
    4. Section 3.3 How has City of Hope “partnered with our Community Practices to activate studies of interest to the community, with a strong focus on Phase II and III studies, early-phase investigator initiated studies and cooperative group trials.”? How has this affected recruitment?The end result has been an increase in accrual rate in community practices where 70-100 patients have been enrolled on therapeutic clinical trials on a yearly basis. Please refer to line 114-115
    5. Response:

  1. Figure 2 the left figure seems to say the patients joining all clinical trials are referred to the main cancer center but the text states trials are being implemented at community centers as well as the main campus.Select trials are run in the community practices and main campus. Many trials have a high level of complexity that does not allow their activation in the community. I appropriate patients are identified in the community practices for these complex trials, and then they are referred to main campus. 
  2. Figure 2 the right figure shows referral of patients to community practices for selected clinical trials.
  3. Response:
  4. Section 5. This is a critical point in this paper. However no specifics in how these treatment pathways are used for metastatic CRC. Line 140 needs a reference to where someone can locate the published NCCN guidelines.Response: Added the URL for NCCN guidelines. Please refer to line 144 and 267.
  5.  

Round 2

Reviewer 1 Report

Dear Editor,

Thank you again for the opportunity to again review "Integrating Academic and Community Practices in the Management of Colorectal Cancer: The City of Hope Model."

After reviewing the authors' revisions and clarifications, I am unfortunately unable to recommend acceptance of this paper due to the premise of the paper being a model for comprehensive colorectal cancer care without providing any evidence of patient or financial benefit.  Initial revisions requested the authors to provide patient outcomes in order to detail how the City of Hope model is beneficial, yet no revision was provided.  This request is crucial as integrating a model for comprehensive cancer care without evidence of effectiveness is potentially clinically and fiscally irresponsible, particularly in the era of quality care reimbursement mandated by the Patient Protection and Affordable Care Act. 

Additionally, the authors did not fulfill the request to specify SEER sources and instead opted for data from the American Cancer Society. SEER is the gold-standard for US cancer registry data from which the ACS gathers data. This is bothersome as it demonstrates the authors' unwillingness to investigate SEER and honor the effort of revision.

Finally, the abstract spends a significant portion of text detailing various genetic mutations, yet the aim of the manuscript was to provide a model for comprehensive colorectal cancer care?  An abstract should funnel towards a specific aim, which in this case was to share a healthcare model that other surrounding community care centers could adopt.  However, the majority of the abstract distracts away from this aim.